# Husserlian Neurophenomenology: Grounding the Anthropology of Experience in Reality

Charles D. Laughlin †

Department of Sociology and Anthropology, Carleton University, Ottawa, ON K1S 5B6, Canada; cdlaughlin@gmail.com; Tel.: +1-206-939-0611
† Current address: 12002 Fremont Ave N, Seattle, WA 98133, USA.

**Abstract:** Anthropology has long resisted becoming a nomothetic science, thus repeatedly missing opportunities to build upon empirical theoretical constructs, choosing instead to back away into a kind of natural history of sociocultural differences. What is required are methods that focus the ethnographic gaze upon the essential structures of perception as well as sociocultural differences. The anthropology of experience and the senses is a recent movement that may be amenable to including a partnership between Husserlian phenomenology and neuroscience to build a framework for evidencing the existence of essential structures of consciousness, and the neurobiological processes that have evolved to present the world to consciousness as adaptively real. The author shows how the amalgamation of essences (sensory objects, relations, horizons, and associated intuitions) and the quest for neural correlates of consciousness can be combined to augment traditional ethnographic research, and thereby nullify the "it's culture all the way down" bias of constructivism.

**Keywords:** scientific anthropology; Edmund Husserl; neurophenomenology; anthropology of the senses; intuition; ethnographic fieldwork; reduction; epoché; collaborative methods

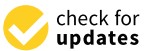



*This is not a "view", an "interpretation" bestowed upon the world. Every view about. . ., every opinion about "the" world, has its ground in the pregiven world. It is from this very ground that I have freed myself through the epoché; I stand above the world, which has now become for me, in a quite peculiar sense, a phenomenon.* [1]

*What, then, does the future hold? In total with the postmodernists, I have come to the regrettable but obvious conclusion that there is no easy accommodation of the scientific and hermeneutic intellectual frames. Since the hermeneutic frame is for me fatally damaged by its denial of objective truth and the possibility of scientific anthropology, my solution is to proclaim that it is not anthropology at all in any reasonable sense of the term. The wave of the anthropological future that I hope for is a scientific anthropology taking into full account the human capacity for discriminating among highly complex combinations of circumstances and reacting systematically to their similarities and differences. Scientific archeology will benefit from such an anthropology, and it will contribute to it in turn. More properly, it will be a part of this anthropology because a properly scientific anthropology searches for significant relationships among all possible sets of variables at all times and places.* [2]

## 1. Introduction

Anthropology has long resisted becoming a nomothetic science. Every time our beloved discipline gets too close to a potential theoretical paradigm, it recoils as from a plague. Anthropologists repeatedly miss any opportunity to build upon empirically relevant theoretical constructs and back away into a kind of natural history of sociocultural differences [2–6]. For a discipline to attain the status of a paradigmatic science in the Kuhnian sense [7]. it must eventually establish its foundations in the structures of reality. Until then, it remains a pre-scientific, naturalistic exploration of the surface of things, an

"artificial" science as Herbert A. Simon [8] characterizes it. Contemporary anthropology is like biology before Darwin, chemistry before the periodic table, or astrophysics before Newton. Anthropology has come close to this paradigmatic Rubicon in the past (e.g., "psychic unity" and Levi-Strauss-style structuralism), only to veer away from fully acknowledging the roots of its scope in the structures of reality—that is to say, the evolutionary biology of the human organism, and the neurophysiology of the brain.

The time is long past since we should be grounding our theory-construction in methods that reveal and study the structures in the real world that produce the similarities and differences between "cultures"; that is, the neural structures that mediate knowledge and operate to confirm that knowledge in direct interaction with the world. One approach that anthropologists have largely ignored is the phenomenology of Edmund Husserl, an approach that provides a rationale and a skillful method for revealing the essential structures of sensory perception [9]. We will see that this method requires the systematic separation of what is given directly by the world to perception and all the cultural information that we automatically layer atop the given. By combining the phenomenology of essences with modern neuroscience, we can then ground anthropological theories—especially those about experience, sensory perception, and embodiment—in research that allows us to discriminate between mental universals and cultural variations. But first let me trace the history of how sociocultural anthropology got to be so methodologically outmoded in the 21st century.

## 2. How Did We Get This Way?

The irony is that the founder of anthropology as a full-on scientific discipline, the great German ethnologist Adolf Bastian (1826–1905), sought an anthropology grounded in the psychophysics of our species see [10–14]. He reasoned, and I think correctly, that the variation in ways of life of different peoples was generated by the same hidden "psychic unity" of the species (read: structure of the human brainmind). It was the task of anthropologists qua ethnographers to perform intensive, long-term, Malinowski-like field research among many traditional peoples (as Bastian himself did for many years) before "civilization" had completely corrupted their lifeways. However, it was not his vision to collect all the weird and wonderful ways that humans carried out their affairs, but rather, by collecting the surface manifestations of variance among many peoples, to inductively produce accurate empirical generalizations that in turn might be deductively explained by theories about the underlying structure—the *Elementargedanken* or "elementary ideas"—operating within the neuropsychology of humankind. Alas, British social anthropology failed to apply Bastian's injunctions and preferred to rely upon second-hand reports from others who had dealings with traditional folk "out in the colonies". Only after the example set by Malinowski's famous exile to the Trobriand Islands did British ethnography turn to participant observation (as opposed to the later American holocultural analysis), and by that point the earlier notions of a structural underpinning of "psychic unity" beneath apparent variation of lifeways were repudiated.

This repudiation of a structural basis of human mentation and behavior was compounded by one of Bastian's students, Franz Boas, who famously migrated to the United States to escape virulent antisemitism in Germany. Boas established himself at Columbia University in New York, and, from that platform, essentially founded American-style "cultural anthropology". From a historical perspective, it is understandable why American cultural anthropology originated in the anti-racist, anti-social Darwinist polemics of the early 20th century [15]. To acknowledge even the slightest biological inheritance of social and cultural traits was to leave open the door to claims of inherent and abhorrent racial characteristics, be they Jewish or African or Aboriginal American. This rather reactionary and scientifically naïve sociopolitical stance, understandable as it might be in hindsight, generates empirically unsupportable claims of the "it's culture all the way down" sort that effectively denies any biogenetic origins of mind, experience, sensation, cognition, evalua-

tion, ethics/morals, or sociality. Everything of interest to the anthropologist is supposedly a cultural or historical construct, the sole result of social learning (e.g., [16–19]).

In the quest for knowledge, whether pre-scientific or scientific, the methods used to acquire knowledge are intimately bound up with the assumptions about reality, and how the truth of things is to be ascertained and embedded in the minds of the curious [7,20]. As the philosopher of science and evaluation Michael Scriven [21] showed us, any agent (human or non-human) that operates as a finite information storage and retrieval system (i.e., a brain) and that finds itself in an over-rich data environment *must theorize*. By theorizing, the agent imposes *redundancies* upon the data field, and thus may adapt to the data field by projecting patterns of redundancy upon the field. All animals with or without brains are just such agents. Among humans, people operate in this way by creating systems of knowledge which, when shared within groups, become belief systems and worldviews. Embedded in such knowledge systems are the means by which the truth of things is ascertained, often by controlling how the quest for truth is accomplished. This relationship between knowledge systems and methods applies as much to scientific as to pre-scientific knowledge systems. Anti-structuralist (or anti-essentialist) social scientists have their views and their methods of confirming their views, as do structuralists and essentialists. But there is a profound difference between the two approaches to knowledge. The anti-structuralist/essentialist views and attendant methods can *never* produce a nomothetic body of theory, whereas the structuralist/essentialist views have a good chance of discovering a theoretical basis for both similarities and differences among human groups. The difference between the approaches is crucial, for one offers no methods by which observations of the surface of things can penetrate into the reality below the surface, whereas the other may well do.

In the past—say, back in Bastian's, Boas' or even Levi-Strauss' day—the technologies available for penetrating to reality were very limited, for each of them were dealing with the products of the human brain operating in groups. There were few methods for directly observing what the brains of people were doing as they expressed themselves, communicated with each other, and behaved cooperatively. Levi-Strauss, for instance, developed a very flawed method of deductive reasoning with respect to observed patterns in social structure and mythology, and only once, to my knowledge, acknowledged that his structures must in fact be neural structures (see his 1971 book *L'Homme nu*, the fourth volume of *Mythologiques*; for the English translation, see [22]).

## 2.1. Experience, Embodiment, and the Senses

Some anthropologists, perhaps weary of the status quo in mainstream sociocultural anthropology, began to ground their understanding of human lifeways upon subjective and intersubjective experience [23–26], including extraordinary or transpersonal experiences [27,28]. With a significant stimulus from earlier theorists such as Durkheim, Levi-Strauss, and Levy-Bruhl [29,30] and the tantalizing cybernetic theorizing of Gregory Bateson [31,32], Victor Turner [23,33] and others began shifting their focus from institutions and social organization (people viewed from the outside, as it were) to the everyday experience of people going about their daily lives (people viewed from the inside). This shift in focus led to the *anthropology of experience* approach to doing ethnography, as well as an understanding of the sociality from an intersubjective standpoint see e.g., [34–38]. The reorientation of perspective towards the lived experience of people in different societies raised interesting methodological issues about how the ethnographer should or could go about "getting into the heads" of their hosts; i.e., how is one to access the privacy of consciousness [39]? This is not a new challenge by any means, but rather requires an expansion of what it means to place "participant observation" into a first person, intersubjective mode [40,41].

The anthropology of experience approach and its quest for appropriate methods led quite naturally to interest in the ideas and methods of phenomenological philosophy [9,30,42–49], especially the formative work of Edmund Husserl [50–52]. The blending

of the anthropology of experience with phenomenological methods led quickly to the recognition, either explicitly or implicitly, that experience is always "embodied"—a realization already well established in early 20th century Husserlian phenomenology [53] (pp. 152–154). *Embodiment* became "a thing" among psychological scientists (see e.g., [54–56]) and became de rigueur within the anthropology of experience cohort [39,49,57–63]. It is easy to demonstrate using even casual introspection that everyday experience is had from an embodied standpoint. I am staring at this screen with my eyes open, from a posture of sitting, and typing with my fingers on a keyboard. Embodiment is as simple and yet as profound as that; as the old saw goes, "wherever you go, there you are", taken there in and by your body, even when you may be "out of body" in a vision or dream.

Yet with this concern with embodiment, with few exceptions (e.g., [64]), scant attention has been paid to the part of the body that mediates experience, namely the nervous system (see [65–68] for exceptions). This persistence of cultural constructivism has gradually alienated anthropology from mainstream post-neuro-turn science [69,70], (see especially [71] on this issue). Even among phenomenological anthropologists, researchers tend to base their findings upon those phenomenologies that support the "culture all the way down" bias (see [42,46,72] for summaries).

Even a cursory reading of Husserl will show that embodiment of experience is fundamental to his understanding of the sensory:

> The Body is, in the first place, the *medium of all perception*: it is the *organ of perception* and is *necessarily* involved in all perception. In seeing, the eyes are directed upon the seen and run over its edges, surfaces, etc. When it touches objects, the hand slides over them. Moving myself, I bring my ear closer to hear. Perceptual apprehension presupposes sensation-contents, which play their necessary role for the constitution of the schemata and, so, for the constitution of the appearances of the real things themselves. [53], (p. 61)

It should be obvious that the anthropology of experience with its understanding of embodiment and its increased reliance upon phenomenological methods would lead to both a sharper focus upon the role of the senses in experience [73–77]—already a primary focus of early 20th century Husserlian methods—and to a closer proximity to psychobiology. Hundreds of books have been written about human social organization, kinship, ritual, behavior, subsistence, and social institutions with no reference at all to people's experiences, their bodies, or their brains. However, it is much harder to avoid human physiology when focusing the ethnographic gaze upon embodied experience and the senses.

## 2.2. Introducing Neuroanthropology

In order to claim, as many anthropologists do, that there are no structural underpinnings mediating apparent individual and cultural variations, logic requires that initially the organ of learning, the brain, be a tabula rasa, a "blank slate" [78]. In other words, the human brain must, by some magic, be at birth like a brand-new, unformatted hard drive. But, in the 21st century, in the age of neuroscience, we know this is not the case [79]. Of course, the emergence of anthropology during the first three quarters of the 20th century occurred during a time when science was burgeoning, but neuroscience was still limited to neurology [80]. Neurologists at the time generally steered clear of addressing the relations between the brain and mind, and few neuroscientists drew any interdisciplinary connections between neurobiology and the social sciences. But, by the 1980s, interdisciplinary neuroscience had begun what we now refer to as the *global neuroscientific turn*, or simply *the neuro-turn*, a neurobiological engagement with research and theoretical issues in the traditional purview of the social sciences and humanities [64,81–83]. This movement led to several new sub-fields in neuroscience resulting in new journals such as *Culture and Brain*, *Social Cognitive and Affective Neuroscience*, and *Social Neuroscience*, as well as numerous related books and articles (e.g., [84–87]).

Biocultural perspectives began to emerge mid-century and may be seen as one early hint of a neuro-turn in anthropology (see [88–90]). Physical anthropology, of course, has had

an abiding interest in the evolution of the primate and hominin brains, especially reflected in the work of Ralph L. Holloway [91,92]. Although the roots of anthropological interest in the brains of fossilized and living peoples dates back to the mid-20th century [93–95], it may be viewed as a more direct outgrowth of *neuroanthropology*, a movement my colleagues and I founded in the 1980s and thereafter elucidated [96–99], in synch with other scientists [100–103], in an attempt to engage cultural anthropology with the neurosciences.

Despite the discipline's aversion to anything suggestive of biological structure, some anthropologists beginning in the 1980s began exploring new approaches to studying human lifeways from an evolutionary biological perspective. The early work leading to the field of *evolutionary psychology* began to appear, spearheaded by Canadian anthropologist Jerome Barkow and his colleagues [104–106]. They reasoned that modern peoples are actually operating from a brainmind that evolved during the Upper Paleolithic, and many "cultural" features we encounter among living peoples make better sense when viewed as ancient adaptations to a physical and social world much changed [107].

Today, neuroanthropology is the study of how the brain mediates experiences, social relations, techno-skills, histories, habits and institutions, and social learning among *Homo sapiens*, and incidentally other large brained social animals, especially other primates, and extinct hominins. We now know that we share many of the neural structures with our fellow creatures with big brains, especially with other primates, and there is much to be learned from cross-species comparisons. As such, neuroanthropology stands as a necessary alternative to the "naïve" culturological position described above—the notion that upon inventing "culture" humanity somehow left their neurobiology behind. Neuroanthropology dispenses with the "it's culture all the way down" fiction and leaves open the possibility of a "cultural" theory grounded upon psychophysical structures (e.g., [68,108–111]).

### 3. Husserlian Phenomenology

As I have suggested above, of all the schools of phenomenology that might have methods to offer us in our quest to ground the scientific study of experience and the senses in reality, the school that promises to be the most applicable is that of Edmund Husserl (1859–1938), considered by many to be the Father of Phenomenology (Those readers wishing to read about Husserl's career, see [112] Chap. 1, [113] Chap. 1, and [51,114]). The reason Husserl's work is advantageous is that, unlike most other phenomenologists such as Heidegger, Merleau-Ponty, and Schutz who have influenced some anthropologists, Husserl was a mature contemplative who showed that to properly reveal and study consciousness via introspection, an extraordinary degree of skill in contemplative methods is required.

Husserl's project is clearly set out in his article "Philosophy as Rigorous Science" [115], in his set of lectures from 1902 to 1903 published in a short book entitled *The Idea of Phenomenology* [116], his book *Ideas: General Introduction to Pure Phenomenology* (commonly called *Ideas I*; [117] (pp. 41–47), and is described as a "new science" that describes and analyses the essential structures of "pure experience" and consciousness. Husserl considered phenomenology to be an *eidetic science*—the term "eidetic" deriving from the Greek *eidos* or "essence" (see [118], Chap. 2). Phenomenology is also an a priori science; that is, it is not "empirical" in the sense of collecting facts from observations had here and there, and testing ideas in light of the facts. Rather, phenomenology is an approach that seeks a first-hand introspective exploration and description of the essential structures of experience, leading to an analysis of essences upon the basis of which scientists can build inductive theories about the world inside and outside the body. Phenomenology is designed not only to precede natural and social science, to ground science in the truth accessible via experience, but also to critique the taken-for-granted grounds of sciences that are, after all is said and done, dependent upon perception.

*3.1. Grounding Sensory Experience in Essential Structures*

Husserl's methods of contemplation are unique. They are also complicated, and there is insufficient space here to do them justice or to show the reader how to perform basic meditations on the senses (see [119–121]). What I do wish to do is emphasize that Husserl's intention was to separate those elements of experience given by the world and those contributed by the mind and projected upon the sensory given. People carry on within their lifeways uncritically blending the two sources of information in what Husserl called the *natural attitude*, the state of consciousness in which we blithely follow our daily habitual engagement with the world of things and people.

> Let us make this clear to ourselves in detail. At the natural standpoint we simply *carry out* all the acts through which the world is there for us. We live naively unreflective in our perceiving and experiencing, in those thetic acts in which the unities of things appear to us, and not only appear but are given with the stamp of "presentness" and "reality". When we pursue natural science, we *carry out* reflexions ordered in accordance with the logic of experience, reflexions in which these realities, given and taken alike, are determined in terms of thought, in which also on the ground of such directly experienced and determined transcendences fresh interfaces are drawn. At the phenomenological standpoinit, acting on lines of general principle, we *tie up* the *performance* of all such cogitative theses, i.e., we "place in brackets" what has been carried out, "we do not associate these theses" with our new inquiries; instead of living *in* them and carrying *them* out, we carry out acts of *reflexion* directed towards them, and these we apprehend as the *absolute* Being which they are. We now live entirely in such acts of the second level, whose datum is the infinite field of absolute experience—*the basic field of Phenomenology*. [117] p. 155

This is why treating Husserlian phenomenology as just another philosophical phenomenology is wrongheaded (see [122]). The intent of Husserlian phenomenology is to strip away all of the taken-for-granted-ness about the world, bundle it all up (metaphorically speaking), and store it in a mental closet until next it is needed, leaving the "pure ego" or Watcher free of beliefs, ontological assumptions, and other hindrances to studying one's own intentional acts.

Husserl variously called his method *performing a reduction* (from the Latin root meaning to "lead back", "bring back", or "restore or return to a previous state"), to *bracket* (from mathematics where an expression in brackets is treated as a unit), to *return to the things* (focus upon what is given in perception and only in perception), to enter the *epoché* (attitude in which taken for granted assumptions about reality are suspended). Simply put, Husserl's method is one of focusing awareness solely upon what is given a priori ("primordially") in sensory experience and suspending all else that is not given in the act [117] (pp. 107–111), [123] Chap. 3, [124] p. 39, [125] (pp. 206–211). Following his teacher Franz Brentano [126], Husserl considered every act of consciousness, whether from the natural standpoint or from within the epoché, as an *intentional* one; that is, consciousness is always about something ("consciousness of"). No matter the state of consciousness, there is always an object and always a subject within the context of the act.

When Husserl performed the reduction upon any sensory experience, he learned that within the field of perception there are objects, relations between objects, a perceptual field, a horizon beyond which no perceptions can occur, and intuitional knowledge. He further discovered, among many other things, that "pure" perception is comprised of sensory qualia (*hyle* or "stuff"), forms (*morphé*), ideas (*eidos*), interrelations, and intuitions that are mediated by *essential structures* (or *essences*) of perception, i.e., in short, the sensuous elements of perception paired with intuitive knowledge. With respect to the object (say a coffee mug), both hyle and morphé interact within the perceptive field to instantiate the idea of the object; the object is not just any random thing, it is a "coffee mug". Moreover, the object/idea relationship is reciprocal. I may see an object and experience it as a "coffee

mug", or I might go looking for a coffee mug (an idea in my head) and find an object that "fulfills" the idea with appropriate hyle and morphé.

Intuition permeates the perceptual act. For instance, we can only perceive the object from a certain perspective, yet we are aware of the object as a whole. We do not perceive a visible half of a coffee mug, but rather the entire coffee cup, visible aspects as well as invisible aspects. Also, we perceive that although the hyle (in this case visual data) can vary with changes in light conditions and changes in perspective, we always intuitively grasp that it is the same object. Husserl himself uses striding towards a tree in his garden as an example:

> The colour of the tree-trunk, as we are aware of it under the conditions of pure perception, is precisely "the same" as that which before the phenomenological reduction we [...] took to be that of the real (*wirklichen*) tree. Now *this* colour, as bracketed, belongs to the noema. But it does not belong to the perceptual experience as a real (*reelles*) integral part of it, although we also find in the experience "a colour-like something", namely, the "sensory colour", the hyletic phase of the concrete experience in which the noematic or "objective" colour "manifests itself *in varying perspectives*". [...] But one and the same noematic colour of which we are thus aware *as* self-same, in itself unchanged within the unity of a continuously changing perceptual consciousness, runs through its perspective variations in a continuous variety of sensory colors. We see a tree unchanged in colour—its own colour as a tree—whilst the positions of the eyes, the relative orientations, change in many respects, the glance wanders ceaselessly over the trunk and branches, whilst we step nearer at the same time, and thus in different ways excite the flow of perceptual experience. Let us now start sensory reflexion, reflexion upon the perspective variations: we apprehend these as self-evident data, and are also able, shifting the standpoint and the direction of attention, to place them with full evidential certainty in relation with the corresponding objective phases, recognize them as corresponding, and thereby also see without further difficulty that the perspective colour-variations, for instance, which belong to some fixed colour of a thing are related to that fixed colour as continuous "variety" is related to "unity". [117] (pp. 283–284)

If you can grasp the distinction between the "objective" color of the tree and the sensory variations we experience through time, you will go a long way toward understanding what Husserl is getting at in general (and, by the way, this is supported by vision neuroscience; see [56] p. 165). What he is saying is that the naïve "natural attitude" observer is experiencing an objective tree defined as having a brown trunk and green leaves, while at the same time glossing over as insignificant the variety of colors that are actually occurring in our sensorium as we change our orientation relative to the tree, and as illumination changes. The role of the hyle is to fulfill the empty eidetic intention "tree". The "objective" tree is the same regardless of the variety of sensory colors we perceive changing through time. We do not have to think about this variety/unity correspondence; it is instantaneous, intuitive, and automatic [127] p. 208.

An essential structure, or essence, is an attribute of the perceptual act that is obdurate relative to our will. In other words, we cannot by any act of will modify the structures of perception. Essences are as it were "wired-in". In neurobiological terms, essences are how we experience the result of millions of years of evolution behind sensory systems that present the world as it is before our sensorium [128]. The essential structures of sensory experience revealed by Husserl's personal meditations are myriad and scattered throughout his writings, most of which have yet to be published (see [129]). Here are several essences that you might uncover by your own efforts as a novice phenomenologist:

**Pattern.** Each and every object you reduce to its primordial given has a form (morphé); i.e., it is ordered, never random, shapeless, or chaotic. Suspending the natural attitude therefore does *not* produce a "blooming, buzzing confusion" of hyle. It is easy to see how the mind apprehends the sensory and intuitive patterns supplied by the primordial given

and project's meaning upon those patterns. You might say that the given is inherently salient within the context of the intentional act.

**Things.** Our experience is full of things, "thingly-real" objects with stable boundaries, and duration. Things are filled with apparent qualia (*hyle* in Husserl's terms as contrasted with form or *morphé*: colors, textures, tones, tastes, etc.), relations among objects, and intuitions (class membership, position within the horizon, aesthetics, pragmatics, and so on).

**Part–whole relationship.** If the object is a three-dimensional thing, we never perceive it in its entirety at the same time. We intuitively infer the whole from the perspectives we are afforded upon its parts. We never "see" part of the coffee mug per se; we see the mug as a whole.

*Eidos* **("form", "type", "species").** Things before our gaze are intuited within direct experience as exemplars of a class, an idea, an *eidos*.

**Entanglement.** Objects (including hyletic data) are always given within the context of an environment of other objects with which they relate and with which they interact within the field.

**Horizon.** Objects always present within the context of a horizon (limits to what can be perceived at the moment). We never perceive the whole world, but only those objects and horizon within our perceptual field.

**Impermanence.** No matter the object of our focus, it is perceptually impermanent. It is not there to your perception before you experience it, and it will not be there at some point in the future. Phenomenologically speaking, there is no such thing as a permanent object, including the ego or Watcher.

**Object–Watcher discrimination.** Reduction of the relationship between the object and the Watcher as the subject. The object only arises before a subject, and the subject intends the object. There is always a subject to any intentional act.

**Intentionality.** Every moment of consciousness is constituted as a system of essential structures linking an object and a subject perceiving the object.

**Attention.** You may exercise control over your relationship with the object, or the object may "draw" your attention. This is the function within consciousness that allows for the willful control of focus and modification of the intentional act.

Many other essences are detectable and describable within the scope of "pure" phenomenological methods. I have only suggested a few to illustrate the flavor and the technique of reducing aspects of perceptual experience. Indeed, once a contemplative is skilled in entering the phenomenological attitude, he can rely upon it in any state of consciousness, including ASC such as psychotropic drug trips, lucid dreams, hypnagogic and hypnopompic states, meditative states, and absorption states. In my own work and writings, I have relied heavily upon both my own experiences enriched by phenomenological methods, and the reports of other mature contemplatives that have mastered these skills, not just those who have mastered Husserl's methods, but from various other traditions, especially Buddhist mindfulness meditators see [130]. Regardless of the approach (as long as it entails intense, skilled, and mature contemplation), reducing the perceptual field always uncovers essences upon which "culture" has no impact save at higher neocortical levels of processing mediating "meaning", which becomes layered on, or "sedimented" upon, the primordial given.

This layering is so automatic and occurs so rapidly that people in general, regardless of their cultural background, are unaware of the processes involved. Only through disciplined phenomenology are we able to make and study these discriminations.

> In the unbroken naïveté in which all psychology, all humanistic disciplines, all human history persists, I, the psychologist, like everyone else, am constantly involved in the performance of self-apperceptions and apperceptions of others. I can of course, in the process thematically reflect upon myself, upon my psychic life and that of others, upon my and others' changing apperceptions; I can also carry out recollections; observingly, with theoretical interest, I can carry out self-perceptions and self-recollections, and through the medium of empathy I can

make use of self-apperceptions of others. I can inquire into my development and that of others; I can thematically pursue history, society's memory, so to speak— but all *such reflection* remains within transcendental naïveté; it is the performance of the transcendental world-apperception which is, so to speak, ready-made, while the transcendental correlate—i.e., the (immediately active or sedimented) functioning intentionality, which is the universal apperception, constitutive of all particular apperceptions, giving them the ontic sense of "psychic experiences (*Erlebnisse*) of this and that human being"—remains completely hidden. [1] p. 209

As Husserl repeatedly noted, culture (including scientific concepts and theories) drapes over our everyday experience like a "garment of ideas" or "garment of symbols" (e.g., [1] p. 51). This, of course, refers to all your life-long *enculturation* ("Enculturation" is anthropology-speak for the process by which we learn to be a "culture-bearing" member of our society)—everything you learn from other people while growing up as a member of a group. Husserl [115–117] argued that only through the application of the phenomenological method could the sciences and philosophy be grounded in direct knowledge about how the mind constructs the reality we experience.

*3.2. Considering Intersubjectivity*

Essential structures of sensory perception extend to objects that happen to be other people, or for that matter, other sentient beings. My perception of you is no different in most respects from my perception of a coffee mug. You are before my consciousness as a thingly-real and enduring object constituted by hyle and morphé, relations to other things in my sensory field, and within my inevitable horizon. All of the essences summarized above apply to people as well as inorganic things. It is in the domain of intuitive knowledge that a crucial difference is discerned. When you appear within my perceptual field, I immediately know you as "like me", with both a body and an inferred stream of consciousness. I was in fact born this way, seeking and finding my mother's face and breasts [79].

My grasp of your consciousness and experience derives from a special type of intentionality Husserl called *empathy* (Ger: *Einfühlung*; [131] p. 120 and p. 135; see also [132–134], [121] p. 92, [135] Chap. 6). This is not to be confused with our everyday, fuzzy notion of empathy that refers to various feelings (compassion, pity, concern, kindness, and the urge to help) or identification with someone's plight [132,136]. My empathy in the phenomenological sense is a generalized intuition associated with my experience of your body, behavior, and expressions [117] p. 210, [124] p. 149; see also [137], [133] p. 11, [138]. We now know that there are systems of so-called *mirror neurons* in the mammalian brain that subserve such intuitions [139]. In other words, I do not just experience your presence as a physical thing, or even a body behaving, but I intuitively know you as a class of "person" and that you, like me, are experiencing a stream of consciousness that I cannot access. "More specifically, what counts in the strict sense as empathy are those experiential acts in which a foreign subject is not merely hypothesized or inferred, but rather given and experienced herself" [132] p. 274. In the same way, when my dog Luke is present to my perception, I experience not only his physical being, but as an exemplar of a class "dog" and that, like myself, he is experiencing a stream of consciousness in which I am an object (see [140]). In neither case can I experience the content of Luke's or your stream of consciousness *in the same primordial way I experience your respective bodies, behaviors, and expressions.*

Empathetic intuition includes the knowledge that the world that I encounter within the epoché is the same knowledge you encounter under the same circumstances. This realization led Husserl to posit one of the most important concepts he contributed to philosophy, as well as to anthropology, social psychology, and other social sciences, the *lifeworld* (Ger.: *Lebenswelt*, Those readers wishing to read more about *lifeworld phenomenology*, see [1]; see also [141–144]). Husserl borrowed the term from philosopher Richard Avenarius (1843–1896) and others to characterize the intersubjectively shared, pre-scientific, primordial givenness of our experience of the environing world and to emphasize the role of intersubjective experience in the performance of the *transcendental epoché*—that is,

realizing that the environing world I am experiencing within the horizon of the primordial sensory given is the same transcendental world you are experiencing via the same essential structures of primordial perception [117] (pp. 130–137), [145].

*3.3. Husserlian Neurophenomenology*

Husserlian essences, whether involved in experiences of things, events, or other sentient beings, are precisely the kind of data that immediately falsify constructivist assumptions, for they not only evidence real structure at the roots of experience, but they are of the kind that can be confirmed by reference to neuroscience, and, in particular, research into the *neural correlates of consciousness* (NCC; see [146–148]). As far as I know, Husserl himself never suggested a merger of his methods with those of neuroscience. Again, his project was the production of a "pure" subjective study upon which to rest the findings of the natural sciences, including psychologies of various sorts [149] (pp. 166–170). However, Husserl was not ignorant of the fact that consciousness and sensory essences were somehow being generated by the brain:

> [I]s it possible, we are asking, for the matter here at issue to be understood in such a way that the cerebral states (states of the [body]) precede, in an Objectively temporal sense, the corresponding conscious lived experiences, or must not, for reasons of principle, the brain state and its conscious accompaniment be simultaneous, in conformity with the absolute sense of simultaneity? Thereby is not a parallelism given *eo ipso*? Namely, in this way: to every conscious lived experience in *my* consciousness $C_m$ there corresponds a certain state in my [body], a certain organic state [read: NCC]. On the other hand, to everything without exception that comprises the [body] there correspond real events of a certain kind in *every* subject, and consequently also in me: certain real perceptual possibilities, which, if not corresponding to the state of the brain [...], then to another state in connection with it in a natural-scientific nexus. [53] p. 305

Husserl raises the question of whether there is a lawful connection (be it simultaneous or over some short "objective" duration) between brain states and states of consciousness. More specifically, do the essential laws of consciousness and experience correspond with laws of psychophysical functions [150]; see also [151,152]? Neuroscience has long since answered Husserl's question in the affirmative. Thus, it now makes sense to speak of *Husserlian neurophenomenology* (HNP) as an interdisciplinary approach to the study of experience and the senses applied to ethnological scopes of inquiry (The phrase "Husserlian neurophenomenology" was coined in an article by neuropsychologist Francisco Varela [153] who was himself a mature contemplative. Varela was influenced by Husserl's writings and seems to have agreed to some extent with Husserlian methods (see [154]). But he contrasted Husserlian neurophenomenology with his own approach which he called "experiential neuroscience". Varela was a practitioner of Tibetan Tantric Buddhism, being a follower of the late Chögyam Trungpa Rinpoche, and the methods Varela used in this regard were different than those used by Husserl. To my knowledge, Varela never applied Husserlian methods in any systematic way).

## 4. Husserlian Neurophenomenology in the Field

In the context of ethnography carried out among traditional societies, unless they have been trained in contemplative phenomenology, our hosts are most likely to be as naïve as is the ethnographer about the essential structures of our individual perceptions. And, like most ethnographers, they will probably be relatively uninformed in neuroscience, although psychophysiological knowledge can be found in some cultural traditions (e.g., see [155] on knowledge of neurophysiology among the Desana people of Brazil). Thus far, there are but a handful of ethnologists who have more than a passing familiarity with either contemplative methods or neuroscience evidence pertaining to NCC. This situation places practical limits on the extent to which most ethnographers can avail themselves of an HNP perspective in the field. But, this merely poses a challenge, not an insurmountable wall, for

to shift anthropology toward a scientific future will undoubtedly require retraining [156]. I would like to conclude this article by exploring some potentially fruitful alternatives that could bring HNP to bear in the field.

### 4.1. Take Up the Challenge

The easiest way to operationalize HNP for any individual ethnographer is to train themselves to apply the reduction before they enter the field. The method is easily teachable. Although it is too much to expect every ethnographer to train in neuroscience as well, this need not be an obstacle because they can collaborate with a willing neuropsychologist. Once a fieldworker is skilled enough in Husserlian phenomenological methods, there are myriad ways to apply these skills in the field. Indeed, this has already been carried out successfully by anthropologists exploring the more esoteric, transpersonal experiences that are the foundation for many traditional societies' spiritual lives and worldviews (e.g., [27,157–159]). It is important to note that Husserlian methods *can be applied in any state of consciousness in which sufficient awareness and lucidity are present*. Indeed, there is a causal connection between the intensity of awareness and intensity of lucidity. The phenomenologically trained ethnographer who subjects themself to native spiritual practices such as drug trips, ritual practices, vision quests, and the like can easily expand their analysis of sensory experiences by applying the reduction. In this way, they can discern the difference between alterations in the sensory given and the meaning attributed to the experiences by their hosts.

### 4.2. Experimental Methods

Experimental ethnography has long been used in the field and in ethnological analysis (see e.g., [160–165]). For instance, volunteer hosts have been trained to use cameras and then asked to photograph and film events of meaning to themselves. Fieldwork in applied anthropology occasionally takes on an experimental expression (e.g., [166]). There is thus no good empirical or ethical reason for not training host volunteers to apply Husserlian methods and then describe their revelations in their native tongues, thus blending Boasian and Husserlian methods for getting at claims of universality for essential structures, and simultaneously exploring how each culture expresses these features of sensory and intersubjective experiences.

An alternative strategy might involve collaboration among professional anthropologists raised in different cultural backgrounds. Native-born anthropologists (Ph.D. level anthropologists today derive from societies in Africa, South and Central America, Native America, First Nations in Canada, Aboriginal Australia, and some islands of the Pacific) and anthropology students could be trained to carry out Husserlian meditations to verify cross-cultural patterns in essential structures and those encountered in alternative state of consciousness (ritual trance, dreaming and co-dreaming, absorption states, states driven by entheogens, ordeals, vision quests, and so on) relevant to their own spiritual traditions.

### 4.3. Training in the Anthropology of the Senses

The most obvious cohort of professional anthropologists that might be (should be?) interested in Husserlian methods are those with a focus upon the *anthropology of the senses* [49,73–77,167–171]. The anthropology of the senses marks a return to "the things" in Husserl-speak, a return to the sensuousness of everyday life [171] (pp. 3–10). Yet, even with the attention being given to sensuousness in experience, *few anthropologists ground their work on the neurobiology of the senses*. Yet, the structure of the sensory systems, as well as the sensorium itself, is as resistant to plasticity as the thyroid, spleen, or intestine; they are some of the most biogenetically ordered neural systems in the body. Offering them a sure-fired method for discriminating the a priori givenness of sense data from cultural overlays of meaning, preferences, and praxis would surely be a welcomed addition to their ethnographic toolkit. There will undoubtedly be resistance from those with an ideologically driven constructivist bias, but enough ethnography could be based upon

Husserlian methods to allow for collaboration with neuroscience colleagues to ground neuroanthropological theories in both ethnographic fieldwork and in neuroscience.

*4.4. Collaborative Teamwork*

As I mentioned, it is rare to find an ethnographer who is also trained in both contemplative phenomenology and neuroscience. Such a broad training would not be strictly necessary, for one of the most powerful applications of Husserlian neurophenomenology in cross-cultural research could be implemented using collaborative teamwork. There is a long history of collaborative ethnography in the discipline. We are all familiar with the expeditions carried out by teams of researchers in the latter 19th and early 20th centuries [172]. Since then, most ethnographic research has been carried out by lone researchers (à la Malinowski) or spouses operating as a team of two (à la the Tedlocks). But cross-disciplinary teamwork has also proved productive in a number of ethnographic venues, including healthcare facilities, ethnic schools in urban areas, market research, linguistic research, as well as more generally in traditional societies (e.g., [173–176]). In fact, collaborative ethnographic approaches and methods are on the rise, and some have suggested that collaborative fieldwork strategies offer distinct advantages, including being conducive to more validity in interpretations, analyses, and theory building [177]. There are now even handbooks and guides to collaborative ethnographic research [178,179], as well as a journal, *Collaborative Anthropologies*.

Hence, there is no good reason not to approach the ethnographic research on the embodiment of experience and the senses using collaborative teamwork that might include neuroscientists, cross-cultural psychologists, ethnographers, and contemplative phenomenologists. The inclusion of a neuropsychologist or a neuroanthropologist and a trained phenomenological ethnographer would seem to be the minimum requirement to assure the application of a Husserlian neurophenomenology.

**5. Conclusions**

Summarizing, I have argued that anthropology can only become a normal science of humanity when it develops methods that can link its theories to the structures of reality. Those structures are the physiological structures of the body and its brain. The shift in the discipline to the study of lived experience and the senses invites the inclusion of Edmund Husserl's contemplative phenomenology and its methods. These methods, which do require training in their application, allow the practitioner to isolate and study the essential structures of consciousness while, at the same time, suspending all the cultural sediments naturally layered atop the primordial given. Combining descriptions of essences with neurobiological research on the neural correlates of consciousness (in other words, Husserlian neurophenomenology) may prove a powerful approach to add to the ethnographer's toolkit.

Operationalizing an ethnographically useful application of Husserlian neurophenomenology may prove challenging, for it, at minimum, requires training (or retraining) the ethnographer's gaze. Ethnographers with training in both contemplative methods and in neuroscience are thin on the ground. It is perhaps impractical at the present time to require an ethnographer to be proficient in both skills, although the time may come when both are routinely taught in anthropological curriculum. I have suggested several approaches, including (1) an ethnographer skilled in Husserlian methods collaborating with a neuropsychologist familiar with the research on NCC; (2) the use of experimental methods with volunteer hosts willing to learn Husserlian methods and then asked to study their sensory experiences and describe in their native language what they discover and what it means to them; and (3) the use of collaborative teamwork in the field so as to bring a number of skills to play, including ethnographic participant observation bolstered by Husserlian methods and neuropsychology.

**Funding:** The research reported in this article received no external funding.

**Informed Consent Statement:** Not applicable.

**Data Availability Statement:** The data referred to in this article are available in the references cited.

**Conflicts of Interest:** The author declares no conflict of interest with respect to this article.

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
