# Peer review of "Husserlian Neurophenomenology: Grounding the Anthropology of Experience in Reality"

_humans, doi:10.3390/humans4010006_

Round 1

Reviewer 1 Report

Comments and Suggestions for Authors

Humans

P 1 L 12-13 --the neurobiological processes that have evolved to present the world to consciousness as it really is.

I think this statement is at best naïve, and most certainly wrong. The world is not presented to consciousness “as it really is” but rather in terms of how our limited human neurosensory system functions, which only can manage a small amount of the information available from the universe. So our sensory system gives us an effective adaptation but not the truth of what the world is really like (i.e, its quantum reality)

P 2 line 87- is ethnology used in a Continental sense rather than US sense (cross-cultural holocultural research) Perhaps should be clarified just what is implied here especially if it is ethnography

P 3 L 106 the truth of things are to be ascertained embedded in the minds of the curious (Kuhn

Seems there should be something between ascertained and embedded

P 4 L 175-1766 alienated anthropology from mainstream post-neuro-turn science

But see a number of anthropologists, including the author, doing just the opposite-- The Supernatural After the Neuro-Turn. New York: Routledge. (ed. Pieter Craffert, John Baker and Michael Winkelman)

P 5 L 123 Ah I see it is cited later (Claffert, Baker and Winkelman where the first editor’s name is misspelled, also in the references-

24. Claffert, PF, Baker JR and Winkelman MJ (eds) 2019. The Supernatural after the Neuro-Turn. New York: Routledge. 684

L473-474-- My empathy in the phenomenological sense is a generalized intuition associated with my experience of your body, behavior, and expressions

Seems to me what is necessary here is a recognition of the innate function of mirror neurons contributing to the fundamental processes by which these intuitions occur

The article presents an approach to anthropology based in Husserlian phenomenology and neuroscience to build a framework to identify the essential structures of consciousness. This approach avoids the excesses of cultural particularism with an eye to identifying the evolved neurobiological processes involved in the presention of the world in consciousness.

The author shows how the amalgamation of essences (sensory objects, relations, horizons, and associated intuitions) and the quest for neural correlates of consciousness can be combined to augment traditional ethnographic research. Anthropology clearly needs new methods to undergird comparative research with an understanding of the structural underpinnings mediating apparent individual and cultural variations.

This article, rather than providing a complete of the review topic of the many schools of phenomenology that might ground the scientific study of experience, has instead focused on the work of Edmund Husserl 253 (1859-1938), appropriately considered the Father of Phenomenology.

The manuscript is well-structured and very well written, and is clearly relevant for the field, with a consider of recent publication beyond the focus on this historical figure (Husserl).

The reasoning of the manuscript is sound and I think the application of the Hussrelian principles are interpreted appropriately. The conclusions show that while it may be challenging, there are ethnographically useful applications of Husserlian neurophenomenology to ethnography.

Author Response

Reviewer #1: Thanks you for the very astute suggestions. I have made appropriate changes for each of your points. Thank you for helping make the piece a bit tighter.

Reviewer 2 Report

Comments and Suggestions for Authors

The article submitted for review presents the highest level of academic research. The Author tackles a very important, extremely difficult and exceptionally complicated problem of grounding the anthropology of experience in reality. The Author conducts the study of this problem not only with great precision and considerable professionalism, but also with extraordinary eloquence. From the outset, the Author clearly suggests an original solution to the research problem posed, which is Husserlian neurophenomenology. The Author thus makes a significant contribution both to the development of research in the social sciences (with particular reference to ethnography) and to the development of the methodology of these sciences themselves. Finally, it is worth mentioning that Husserl himself was to reiterate during his lifetime that he was merely paving the way for future generations of researchers who would unveil the proper potential hidden in the phenomenological method. The article discussed here certainly fulfils this dream of Husserl's. In his last lectures, Husserl also floated a vision of the crisis of the European sciences, the solution to which was to be found in phenomenology. The reading of the text under review seems to confirm this intuition of the founder of phenomenology: it can certainly solve many crises both in the social sciences and in ethnography. 

Finally, it is worth mentioning that many contemporary scientists, for various reasons, approach many centuries of philosophical achievements in the areas of epistemology, metaphysics, ontology and anthropology with a considerable degree of scepticism. Some even ridicule philosophy as a manifestation of the naivety of the bygone and supposedly "dark ages", which are nowhere near our enlightened era of science. However, as the Author of this paper clearly demonstrates, many modern sciences can benefit immeasurably from the inspirations drawn from the work of the great philosophers and thinkers of the past.

In conclusion, I would like to applaud and congratulate the Author on the publication of a significant article for the development of research in anthropology.

Author Response

Reviewer #2: Thank you for your support and and approval for my argument.